# Inventive nesting behaviour in the keyhole wasp *Pachodynerus nasidens* Latreille (Hymenoptera: Vespidae) in Australia, and the risk to aviation safety

**Alan P. N. House****¹\*, Jackson G. Ring², Phillip P. Shaw³**

**1** Eco Logical Australia, Brisbane, Queensland, Australia, **2** Operations Section, Brisbane Airport Corporation, Brisbane, Queensland, Australia, **3** Ecosure Pty Ltd, Burleigh Heads, Queensland, Australia

\* alan.house@ecoaus.com.au

## Abstract

The keyhole wasp (*Pachodynerus nasidens* Latreille 1812), a mud-nesting wasp native to South and Central America and the Caribbean, is a relatively recent (2010) arrival in Australia. In its native range it is known to use man-made cavities to construct nests. A series of serious safety incidents Brisbane Airport related to the obstruction of vital airspeed measuring pitot probes on aircraft possibly caused by mud-nesting wasps, prompted an assessment of risk. An experiment was designed to determine the species responsible, the types of aircraft most affected, the seasonal pattern of potential risk and the spatial distribution of risk on the airport. A series of replica pitot probes were constructed using 3D-printing technology, representing aircraft with high numbers of movements (landings and take-offs), and mounted at four locations at the airport. Probes were monitored for 39 months. Probes blocked by mud nesting wasps were retrieved and incubated in mesh bags. Emerging wasps were identified to species. Results show that all nests in probes were made by *P. nasidens*, and peak nesting occurs in the summer months. Nesting success (as proportion of nests with live adult emergents) was optimal between 24 and 31°C and that probes with apertures of more than 3 mm diameter are preferred. Not all areas on the airport are affected equally, with the majority of nests constructed in one area. The proportion of grassed areas within 1000 m of probes was a significant predictor of nesting, and probe volume may determine the sex of emerging wasps.

## Introduction

Interactions between aircraft and wildlife are frequent and can have serious financial and safety consequences. Birds are the most common threat to aircraft [1, 2], with a host of terrestrial animals also implicated [3]. There were over 16,500 reported incidents involving birds and a further 397 involving other vertebrates at Australian airfields between 2008 and 2017 [4]. The majority of these incidents occurred during take-off (23.45%) or landing (34.24%), the two most vulnerable phases of flight. However, the risk posed by wildlife when aircraft are

**Data Availability Statement:** All data files are available from the OSF database (https://doi.org/10.17605/OSF.IO/T87ZY).

**Funding:** The funder (BAC) provided support in the form of salaries for authors [APNH, PPS, JGR] and provided access to the airport to conduct the experiment, co-designed the experiment and assisted in monitoring. Qantas (Matthew Hill is a paid employee of Qantas) made significant in-kind (but not financial) contributions to the project through professional advice and access to materials and 3D-printing facilities. the funders did not have any additional role in the study design, data collection and analysis, decision to publish, or preparation of the manuscript. The specific roles of these authors are articulated in the 'author contributions' section.

**Competing interests:** This work was commissioned and funded in its entirety by Brisbane Airport Corporation (JGR is a paid employee of BAC) to address a serious threat to aviation safety at the airport. No other sources of funding were received by the project team. BAC provided salaries to JGR, APNH and PPS. Ecosure (PPS) and Eco Logical Australia (APNH) (both consultancies) were engaged to provide professional ecological advice and devise and run a research program. Qantas (MJH is a paid employee of Qantas) made significant in-kind (but not financial) contributions to the project through professional advice and access to materials and 3D-printing facilities. There are no financial competing interests beyond these relationships. As the funder of the work, BAC (through JGR) provided access to the airport to conduct the experiment, co-designed the experiment and assisted in monitoring. All analysis and interpretation was done by Alan House. This does not alter our adherence to PLoS ONE policies on sharing data and materials.

on the ground is much less understood and despite widespread anecdotal knowledge amongst the tropical and sub-tropical aviation community about insects and flight safety, specific threats posed by insects have not been quantified before. Here we report on an emerging insect-aircraft interaction that is a potentially lethal threat to flight safety. The consequences and costs of not understanding this threat could be catastrophic [5].

In 2012, an exotic mud-nesting wasp (the keyhole wasp, *Pachodynerus nasidens* Latreille (Vespidae: Eumeninae)) was detected at Brisbane Airport (caught in an office off the international arrivals hall: Ross Wylie pers. comm.). This was the second record of the species in Australia after one was detected in northern Brisbane during a routine quarantine inspection of cargo initially received at the Port of Brisbane in 2010. *Pachodynerus nasidens* is native to tropical Central and South America and the Caribbean, and is also found in the southern USA (Florida, Texas and Arizona where it is possibly adventive [6, 7]). It has also been recorded from a number of Pacific islands including Hawaii, Polynesia, Micronesia and Japan [8, 9].

*Pachodynerus nasidens* is well known as an inquiline species, using abandoned or empty nests of other wasps, and for using man-made cavities (e.g. window crevices, keyholes, electrical sockets), to the extent that wholly constructed nests by the species are rare [10]. Indeed, it is known in USA as the "keyhole wasp". It will also nest in the ground and construct mud nests attached to plants: this plasticity in nesting behaviour is remarkable [6] and allows the species to adapt to new opportunities for nesting in novel environments. It is a relatively small wasp, distinguished from other species in the genus by the relatively short thorax and dull yellow-brown markings (as opposed to bright yellow) on distal abdominal tergites [11].

In its native range it is found in grasslands and swamps dominated by grasses and sedges [12]. It is found at similar latitudes in South America (e.g. Guarapuava, Brazil) as in Brisbane, from high altitudes (1120 m) in climates with cool wet seasons and frosts to sea level [12].

## The risk to flight safety

Airspeed is a critically important measure in aviation. Without an accurate measure of airspeed pilots cannot easily judge take-off and landing and cannot ensure that the aircraft flies in a safe speed range; fast enough to generate lift and remain airborne but also slow enough to avoid structural failure under excessive aerodynamic loads.

Pitot probes are hollow tube-like instruments that are commonly mounted on the fuselage behind the nose cone and below to the cockpit. The probes measure airspeed by subtracting mechanically the ambient "static pressure", sampled by a separate sheltered hollow static port, from the "total pressure" of air entering the exposed pitot probe to give a "dynamic pressure". Dynamic pressure directly equates to speed through the air.

Aircraft manufacturers and aviation authorities recognise that amongst the issues that might cause the pitot static system to fail to indicate correct airspeed, a blockage (or even partial blockage) of a pitot probe or static port by insects (or other agents such as ice) is a significant hazard [13–19]. Anomalies between separate airspeed indicators can lead to costly and hazardous rejected take-offs or turn backs [e.g. 16, 17] or even catastrophic consequences [13, 14, 18, 19]. In February 1996 a Boeing 757 crashed shortly after take-off from the Dominican Republic, killing all 189 passengers and crew. Anomalous airspeed readings from the pitot probes were responsible for the pilots misjudging the aircraft's speed. A sphecid (mud-dauber) wasp was believed to have made a nest in one of the pitot probes, although none were recovered [20]. The plane had been standing at Gregorio Luperón International Airport in Puerto Plata, and for the two days prior to the fateful flight the pitot probes were not covered as recommended by the manufacturer [14].

At Brisbane Airport, five incidents were reported between January and March 2006 in which pitot probes gave inconsistent readings and flights were either rejected on take-off or proceeded to their destination. All aircraft were Airbus A330s. An incident on 19 March 2006 led to dangerous brake heating and tyre deflation when take-off was rejected, resulting in the deployment of airport rescue and fire-fighting services [16]. On inspection, it was found that the pitot probe on the pilot-in-command's side had "wasp-related debris". Material retrieved from this probe by the Australian Museum included fragments of an insect body including the head of a wasp.

In November 2013, an A330 prepared to take off but returned to the bay after airspeed discrepancies between the captain's and first officer's readings. After checking electronic instrumentation the plane was cleared for a second take-off, but another airspeed discrepancy occurred, and the captain returned under emergency procedures. Sand and mud consistent with a mud dauber wasp nest was found to be blocking the captain's probe [17]. Since this incident, more detailed records of wasp-related issues have been recorded at Brisbane and a total of 26 were reported between November 2013 and April 2019.

Partial completion of a nest in a pitot probe can take place very quickly. An A320 was found to have a blocked total air temperature probe on arrival in Newcastle from Brisbane in August 2015. The aircraft had been on the ground in Brisbane for only 30 minutes.

This paper reports on research at Brisbane Airport to answer some simple questions that could be addressed by a field experiment, and which could give clear information to inform a management response: What species were causing pitot blockages? Where on the airport is most likely to be impacted? What aircraft types are most likely to be affected, and what aspects of pitot probes might determine this? And what seasonal and environmental attributes contribute to temporal and spatial patterns of blockage? Data on nesting preferences, seasonality, and reproductive success will be essential in directing management to reduce the risks posed by this threat to airlines and the flying public.

Brisbane Airport Corporation, as owners of the airport, gave permission to conduct this work. No further permits or approvals were necessary. No ethics approval was required under Australian law to conduct this research.

## Materials and methods

### Study site

Brisbane is located on the eastern seaboard of Australia (27.3911˚S, 153.1197˚E), on the shores of Moreton Bay, Queensland at approximately 4 m above sea level. It is flanked on the east by the Brisbane River, and on the west by ecologically significant coastal wetlands (mangroves and saltmarshes) and to the south by the city of Brisbane. The climate is sub-tropical, with a long-term mean annual rainfall of 1190 mm, 50% of which falls between December and March. The annual mean maximum temperature is 25.4˚C, and annual mean minimum is 15.7˚C, with fewer than 2 frost days per year [21].

### Probe deployment

Replica pitot probes were installed at Brisbane Airport between February 2016 and April 2019. Pitot probes vary in design between each airframe type, so aircraft movement data (frequency of arrivals and departures) from 2015 at Brisbane Airport and aircraft location (which terminals and gates aircraft used) were used to determine which airframes should be included in the study. Airframes were also filtered for pitot probe dimensions, so that a range of aperture diameters and chamber depths (i.e. distance to first baffle) were used. Six airframes/pitot probe types were selected (Table 1).

                                                                            

**Table 1. Dimensions of pitot probes for six airframes types and deployment locations at Brisbane Airport.**

| Airframe | Aperture diameter (mm) | Distance to first baffle (mm) | Volume (mm³) | Tip form | Locations |
|---|---|---|---|---|---|
| Embraer ERJ90 | 2.5 | 29.0 | 570 | Concave | QLINK, DTBN, DTBS |
| De Havilland Dash 8 | 4.0 | 19.0 | 239 | Concave | QLINK, DTBN, DTBS |
| Boeing 737–800 | 5.0 | 62.0 | 1218 | Concave | QLINK, DTBN, DTBS, ITB |
| Airbus A330 | 5.5 | 33.0 | 784 | Concave | QLINK, DTBN, DTBS, ITB |
| Boeing 737–400 | 7.2 (9/32") | 33.0 | 1344 | Convex | QLINK, ITB |
| Boeing 747–400 | 8 (5/16") | 36.0 | 1810 | Convex | QLINK, DTBN, DTBS, ITB |

QLINK, QantasLink apron; DTBN, Domestic Terminal North; DTBS, Domestic Terminal South; ITB, International Terminal (see Fig 2).

As sufficient decommissioned pitot probes were difficult to source, we manufactured probes from UV-resistant ABS plastic using 3-D printing technology, based on detailed engineering designs for accuracy. Probes were mounted on 3 mm x 900 mm x 1200 mm white-painted steel sheets (to simulate an aircraft fuselage) which were themselves attached to a weld-mesh panel for rigidity (Fig 1). Panels were secured to structures as close as possible to aircraft parking locations (e.g. gate light poles at the domestic terminal, aerobridges at the international terminal; Fig 1) with packaging straps and heavy duty stretch straps and flagged for visibility. They were erected at heights similar to the aircraft-mounted probes (2–2.5 m at the domestic terminal for e.g. A320, 4 m at the international terminal for e.g. 747–400 aircraft) (Fig 1).

Three panels were established at each of four locations at Brisbane Airport (Fig 2) where incidents involving potential mud wasp activity was suspected or where wasps had been recorded as potentially threatening aircraft: the northern domestic terminal (QLINK and DTBN), southern domestic terminal (DTBS) and the international terminal (ITB). The minimum distance between groups of three panels was 233 m.

## Intercept traps

In addition to the mounted replica probes at the gates, a series of intercept traps were established to determine if nesting in pitot probes could be reduced by attracting females to

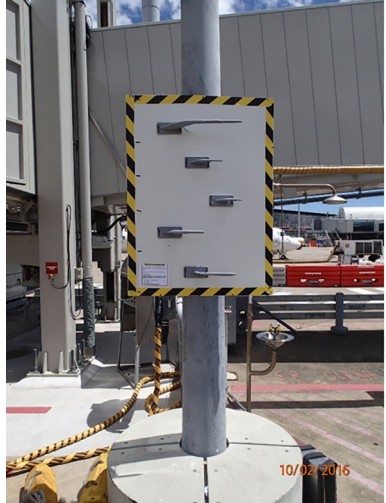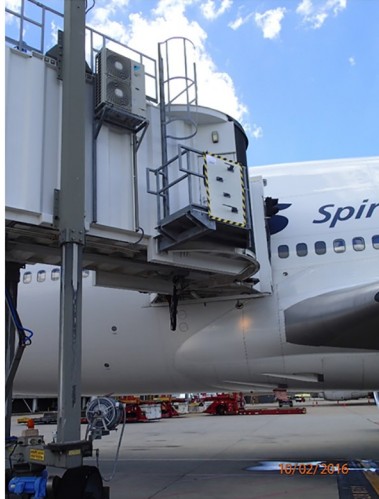

**Fig 1.** Pitot panels installed on gate light pole at domestic terminal (left) and on aerobridge at international terminal (right).

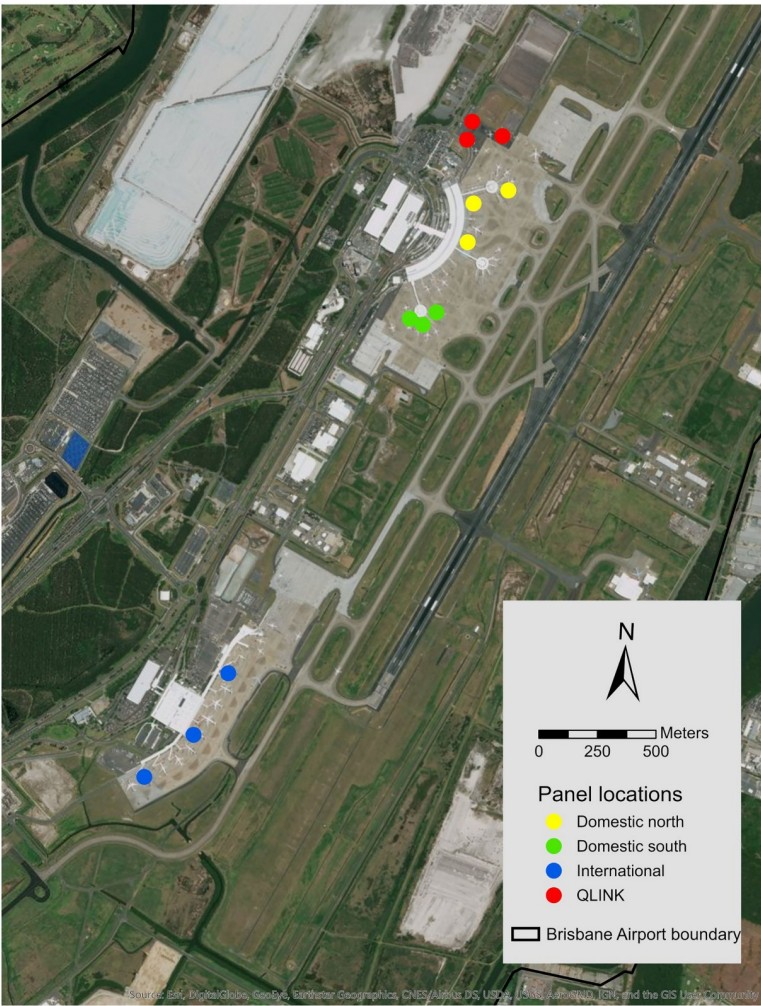

**Fig 2. Location of pitot probe panels at Brisbane Airport.** Created using ArcGIS® software by Esri. ArcGIS® and ArcMap™ are the intellectual property of Esri and are used herein under license. Copyright © Esri. All rights reserved. For more information about Esri® software, please visit www.esri.com.

alternative nest sites away from stationary aircraft. These were established in November 2018 and monitored weekly until April 2019 in both airside (16 traps) and landside (11 traps) locations. Each trap consisted of 68 cardboard tubes 8 mm in diameter and 153 mm long (total of 1088 airside tubes, 748 landside), with one open end. Blocked tubes were removed and replaced by clean ones at weekly intervals.

## Monitoring

All probes were inspected weekly to monthly, depending on season and wasp activity, between February 2016 and April 2019: a total of 49 inspections were made between these dates. Blocked probes were removed from the panels and placed in fine mesh bags with labels recording date, probe type and location, and replaced with empty probes of the same type. Bags were kept in a mildly air-conditioned room (25–27˚C) and monitored for emergence. Emerging invertebrates were placed in vials with methanol, and the date, species, number of individuals and sex were recorded. Species that could not be identified were sent to the Queensland

**Table 2. Broad habitat types mapped at Brisbane Airport.**

| Habitat | Description | Value to wasps | Location on airport |
|---|---|---|---|
| Natural horizontal | Predominantly grassed areas e.g. around runways, taxiways, aprons | Source of prey items; water and sediment for nesting | Airside, landside |
| Natural vertical | Treed areas e.g. *Casuarina* plantations | Source of prey items; water and sediment for nesting | Landside |
| Mixed natural | Tree/shrub/ground cover mixtures e.g. garden areas around landside infrastructure | Source of prey items; water and sediment for nesting | Landside |
| Built horizontal | Bitumen or concreted areas e.g. runways, taxiways, major roads (includes construction sites) | Sediment and water for nests | Airside, landside |

Museum for determination. Retrieved probes that remained blocked after 70 days were examined for remains of undeveloped wasps and unconsumed prey.

Broad habitat types that wasps might respond to and be influenced by in terms of resource availability (i.e. water and sediment for nest building, vegetation for Lepidopteran prey) were mapped around each pitot panel location using Nearmaps® satellite imagery from May 2016. Habitats were mapped into the classes in Table 2, and the areas of each habitat class calculated for 20 m, 50 m, 100 m, 200 m, 500 m and 1000 m radii around each panel in ArcGIS. Multiple regression analyses were used to investigate relationships between the number of blockages (nesting) and the proportions of habitat classes in the environment.

## Data analysis

We analysed the wasps' probe choice, blockage rates, locations and the relationship between nesting and environment. As we did not have a perfectly balanced design (in respect of numbers of probes of each airframe), probe blockage rates were expressed as blockages per 12 probes for analyses of probe choice. Generalised linear mixed models were used in R [22] to explore relationships between blocked probes and location and probe type for probes that were deployed at all locations (A330, 737–800, 747–400). In these, the number of blocked probes was the fixed effect and random variables location and probe type. The Bernoulli distribution was used (probes were classed as blocked or not blocked), with a Monte Carlo simulation size of $10^4$ to achieve stable maximum likelihood estimates [22]. Multiple linear regression was used to assess the influence of climatic variables (rainfall in month of blockage, rainfall in previous 3 months, mean annual maximum and minimum temperatures: all climatic data taken from Brisbane Aero Station [21]) and spatial environment on blockages. Analysis of variance was performed on blockage rates and pitot probe diameter and volume, and a Cochran-Armitage trend test [23] applied to sex ratios of emergent wasps and probe types. Study questions and associated tests, figure and tables are listed in Table 3.

**Table 3. Question posed and tests applied in this study.**

| Questions | Tests | Figures | Tables |
|---|---|---|---|
| Is there a location preference for nesting? | Generalised linear mixed model | - | - |
| Is there a pitot probe preference for nesting? | Generalised linear mixed model ANOVA | - | Table 4 |
| What is the seasonality of nesting? | ANOVA | Fig 3 | - |
| What are the environmental factors influencing nesting location? | Multiple linear regression | - | Table 5 |
| What is the seasonality of nesting success (i.e. emergence)? | ANOVA | Fig 4 | - |
| Are sex ratios of emerging wasps influenced by pitot probe type? | Cochran-Armitage trend test Linear regression | Fig 5 | - |

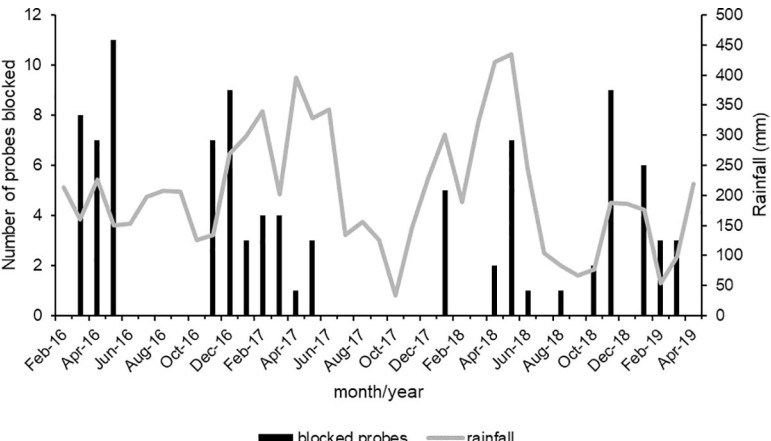

**Fig 3. Pitot probe blockages by mud-nesting wasps and rainfall at Brisbane Airport, February 2016 –April 2019.**

## Results

### Seasonality of pitot blockages

The first pitot probe blockages occurred two weeks after the probe panels were installed. In total, 93 instances of fully blocked probes were recorded during the 39-month study. There is a seasonal pattern, with 96% of nesting occurring between November and May (Fig 3). Data from 2017 indicate that nesting of *P. nasidens* extended into July, with hatching as late as August. Nesting has been recorded in every month of the year except July and September. There is no relationship between nesting and rainfall in the month of nesting ($F_{1,37} = 0.0001$, $P = 0.99$), or with rainfall in the previous 3 months ($F_{1,37} = 0.837$, $P = 0.37$). However, nesting was significantly correlated with the average monthly temperature ($F_{1,37} = 7.33$, $P = 0.01$).

### Probe types blocked and location on airport

The blockage rate by aircraft type indicate that probes with apertures $\geq$ 2.5 mm are preferred (Table 4). Proportionally, probes from 737 airframes (737–400 and 737–800, diameters 5–7.2 mm) are most likely to be blocked (56.3% of all blockages), especially the 400 series, followed by the A330 (19.3%, diameter 5.5 mm). The smallest probe (Embraer ERJ90, diameter 2.5 mm) had the lowest rate of blockage. However, neither probe aperture diameter nor distance to first baffle were significant predictors of nest choice ($F_{2,57} = 0.909$, $P > 0.05$).

The mixed effects model confirmed that location was a highly significant predictor of nesting ($z = -7.942$, $P < 0.0001$), whereas probe type was not ($z = 1.205$, $P > 0.05$) for the three probe types deployed at all locations (A330, 737–800, 747–400): however, when all probes are

**Table 4. Pitot blockages by aircraft type, February 2016-April 2019 at Brisbane Airport.**

| Airframe type | Standardised rate of blockages (per 12 probes) | Probe aperture diameter (mm) |
|---|---|---|
| ERJ90 | 1.3 | 2.5 |
| DHC-8 | 14.6 | 4.0 |
| B737-800 | 23 | 5.0 |
| A330 | 23 | 5.5 |
| B737-400 | 44 | 7.2 |
| B747-400 | 13 | 8 |

considered across all locations, 86% of blockages occurred in probes with a diameter range of 5–8 mm.

In addition to this data on nesting in pitot probes, *P. nasidens* was recorded nesting in a number of other locations at the airport including on-ground servicing equipment, and in intercept trap nests that were separately deployed to trap wasps.

## Habitat distributions

Probes closer to natural habitats are more likely to be blocked than those further away: linear regression between the number of blockages and the minimum distance of panels to each habitat type shows that the proximity to natural horizontal habitat is significant (adjusted $R^2$ = 0.49, $F_{1,10}$ = 11.41, P = 0.007) but proximity to natural vertical habitat is not (adjusted $R^2$ = -0.099, $F_{1,10}$ = 0.012, P = 0.915). A multiple regression of the number of probe blockages and the proportion of habitat types at increasing distance shows that the amount of natural horizontal habitat at 1000 m is also significant (Table 5).

## Nesting success

There were 93 occurrences of blocked pitot probes. Of these, 37 (39.8%) produced live adult wasps, 18 (19.4%) had developed but unhatched wasp imagos, and 38 (40.9%) had contents that were either undeveloped or parasitised. All adult mud-nesting wasps that emerged from pitot probes were *P. nasidens*.

There is no consistent trend in successful nesting (i.e. completed nests and live hatching) with time of year (Fig 4), and no significant relationship between hatching and rainfall during nesting ($F_{1,37}$ = 0.010, P >> 0.05): however, rainfall during the previous 3 months and nesting success was positively related ($F_{1,37}$ = 7.998, P < 0.01). Similarly, mean maximum temperature during nest development and ultimate nesting success were not related ($F_{1,37}$ = 0.409, P > 0.05). Probes blocked during the peak of summer temperatures and rainfall (January-March: man max. temp. over 3 years 29.6˚C) had lower success rates, nests completed late in the nesting season (April-June: mean max. temp. 24.1˚C) had a greater likelihood of successful emergence and probes blocked after mid-May-October (mean max. temp. 23.4˚C) developed to the adult stage but did not hatch.

Incubation times varied greatly, from 16 to 138 days with an average of 45. In the 2016–17 and 2017–18 summer nesting periods, nests that were completed at the start and the end of these periods appear to mature more quickly (Fig 4).

**Table 5. Multiple regression analysis of proportions of each broad habitat type at increasing distance from probe panels and rate of probe blockage.**

| | | | | | | | | | | | | | | |
|---|---|---|---|---|---|---|---|---|---|---|---|---|---|---|
| | | | Habitat | | | | | | | | | | | |
| | ANOVA | | natural horizontal | | natural vertical | | mixed natural | | built horizontal | | built vertical | | water | |
| Distance (m) | F | P | t | P | t | P | t | P | t | P | t | P | t | P |
| 20 | 2.343 | 0.157 | - | - | - | - | - | - | -1.531 | 0.157 | - | - | - | - |
| 50 | 4.849 | **0.037** | - | - | - | - | - | - | -1.345 | 0.211 | -2.809 | *0.020* | - | - |
| 100 | 2.913 | 0.101 | 0.122 | 0.913 | - | - | - | - | 0.091 | 0.930 | 0.074 | 0.943 | - | - |
| 200 | 7.782 | **0.010** | -2.166 | 0.067 | - | - | -3.340 | *0.012* | -2.610 | *0.035* | -2.836 | *0.025* | - | - |
| 500 | 2.402 | 0.177 | 1.425 | 0.214 | 1.429 | 0.212 | 1.399 | 0.221 | 1.423 | 0.214 | 1.401 | 0.220 | 0.921 | 0.399 |
| 1000 | 5.399 | **0.032** | 2.461 | **0.049** | 2.305 | 0.061 | -1.637 | 0.153 | 2.229 | 0.067 | 2.191 | 0.071 | -0.887 | 0.395 |

P values in bold indicate positive significant relationship; P values in italics indicate significant negative relationship.

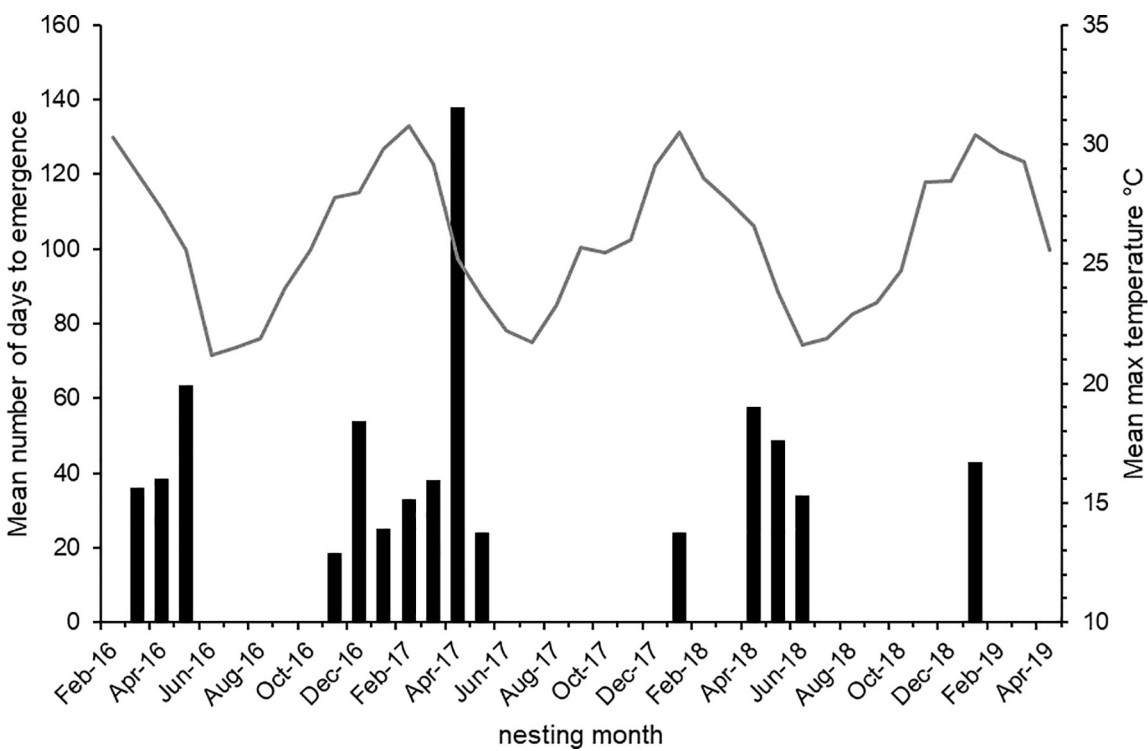

**Fig 4. Mean number of days (from collection of probes) to emergence of live wasps, February 2016 –April 2019.**

Blocked probes have had 1–3 cells built, but only two (both A330), produced two adults, and one produced three (B737-800). There was no significant difference in probe types in the number of cells yielding more than one adult wasp ($F_{4,34} = 0.53$, P > 0.05).

Probes that did not result in successful hatching were examined for remains of undeveloped wasps and unconsumed prey. All had either pre-adult *P. nasidens* pupae or had evidence of caterpillar and/or beetle larvae: none had evidence of other native wasps or spiders (which would indicate other genera of Vespidae).

Only one probe produced live adults of a potential parasitoid of *P. nasidens*: five *Chrysis lincea* (Chrysididae) adults hatched from a B737-400 probe retrieved in May 2017.

## Sex ratios

A male-biased sex ratio of 1.53:1 was recorded from the pitot probes: this does not significantly deviate from a 1:1 relationship (Cochran-Armitage z = 1.042, P = 0.2975). Males and females were equally likely to hatch from A330 and 747–800 probes (observed ratio 1:1), but males were 3 and 1.75 times more likely to hatch from 737–800 and 737–400 probes respectively. There is also a close curvilinear relationship ($R^2 = 0.942$, P = 0.027) between the proportion of hatching males and pitot probe volume with the greatest proportion hatching from mid-sized probes, but this relationship is not as close for females ($R^2 = 0.517$, P = 0.372) (Fig 5). A test for outliers in this dataset returned none.

## Discussion

Nesting by *P. nasidens* at Brisbane Airport follows general seasonal patterns found within its native range [24], with peak nesting occurring in warmer, wetter months although the only

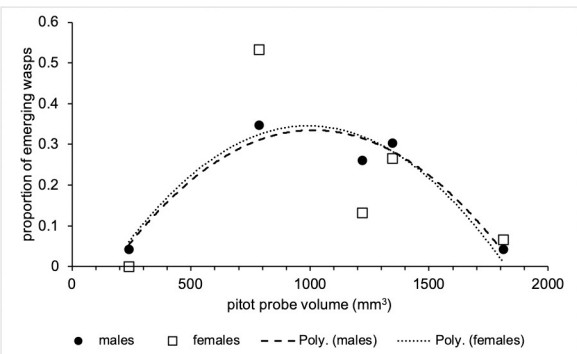

**Fig 5. Relationship between pitot probe volume and sex of emerging live adult wasps.** Second order polynomial line of best fit fitted. $R^2_{male} = 0.942$ (p < 0.05), $R^2_{female} = 0.517$ (ns).

significant relationship was to temperature. Nesting success (as expressed by the proportion of nests producing live adults) is very consistent with other studies of this species (39.8% compared with 39.7% in [25], and incubation times are also generally within ranges published for native populations (16–138 days compared with 23–41 days [25], 5–59 days [24], 20–24 days [26].

Nesting activity is mostly confined to the summer months, which broadly agrees with other studies [24, 26], where egg development was optimal between 26–31˚C. At Brisbane, nests completed after the peak of summer (when mean maximum temperatures fell below 26˚C) still developed and hatched adults, suggesting some local climatic adaptation to slightly lower temperatures.

Sex ratios of *P. nasidens* emerging from trap nests within its native range are variable, from 2:1 male to female (from a very small sample size) [12], 1:1 [27], and 0.56:1 [24]. Evidence suggests that local conditions of weather, prey availability and nest opportunities can affect sex ratios in cavity-nesting species [28, 29]. Data from this study suggests that there may be an effect of pitot probe nest volume on the proportion of males successfully hatching, but not the number of females.

The emergence of *Chrysis lincea* from blocked pitot probes indicates that some parasitism of *P. nasidens* is occurring, and the presence of *Melittobia* sp., *Cotesia* sp. and *Perilampus* sp. from mud nests on adjacent terminal buildings suggests that these parasitoids may also contribute to control of *P. nasidens*. A *Melittobia* species parasitised *P. nasidens* in Cuba, affecting between 41.6 and 75% of all cells [30].

The occurrence of *P. nasidens* at Brisbane Airport and the high number of pitot incidents there due to mud wasps indicates that this species is primarily responsible, but other species cannot be excluded. The earliest recorded aircraft incident in 2006, if attributable to this species, indicates that it may have arrived some time before the first official record in north Brisbane 2010 and at Brisbane Airport in 2012. Other mud nesting wasp species known to be present at the airport include (from observation and live captures) include *Abispa ephippium*, *Anterhynchium nigricinctum*, *Delta campaniforme*, *Delta* sp. aff. *nigricornis*, *Delta* sp. (Vespidae: Eumeninae), *Chalybion* sp. aff. *bengalense*, *Sceliphron formosum* (Sphecidae: Sceliphrinae), *Pison pyrrhicum*, *Pison* sp. (Crabronidae: Crabroninae).

The choice of pitot probe type for nesting in *P. nasidens* appears to be based on a minimum aperture of 2.5–3 mm. Adult females are 10–12 mm long, and observations of active nesting in probes showed that wasps leave head-first, indicating that they turn around inside the probe.

A smaller aperture may be enough to allow a pupa to develop and hatch, but adult females cannot physically negotiate the narrow bore. In Brazil, *P. nasidens* was found to prefer apertures of 6–9 mm and did not use traps with apertures of 4 mm which were also available [11], and *P. nasidens* and its congeners did not nest in cavities with apertures above 7 mm in diameter [24]. However, *P. nasidens* has been found nesting in bamboo canes between 17 and 23 mm in diameter [31]. Of the other mud wasps at Brisbane Airport (above), only *Pison pyrrhicum* is likely to be small enough (7.4–7.7 mm long) to negotiate the pitot probes. Females of the other species are 18 mm (*Chalybion bengalense*) to 30 mm (*Abispa ephippium*) in length.

The environment available at Brisbane Airport to *P. nasidens* for securing nesting resources is entirely modified. All grassed areas are managed (i.e. mown) to reduce their attraction to birds; plantings of native species in plantations and gardens are similarly semi-natural at best. The closest environments to the pitot probe panels are highly modified structures that provide few nesting resources (prey, sediment and water), except for pitot probes. There is evidence that the extent of grassy habitats at 1000 m from the panels may influence nesting success: this shows that wasps are prepared to use the pitot probes despite making longer flights to gather nest building resources.

Nesting in this species is extremely efficient at Brisbane Airport. Time to complete a two-cell nest may be as short as a few hours: anecdotal evidence suggests that a complete nest can be constructed in a pitot probe within 5 hours–this occurred at night between arrival at 20:41 and 05:49 on 1 March 2014 (Boeing 737). In its native range (i.e. Jamaica) the total time to complete a cell is between 2.5 and 4.75 hrs, with a 3-cell nest in a 6 mm diameter cavity completed in 3.5 hrs [26]. As most nests in pitot probes at Brisbane are only single-celled, nest-building is considerably more rapid and may be limited to provisioning a single cell and applying a closing plug of mud. In respect of dangers to aircraft however, the nest does not need to be complete: the first addition of mud for the rearmost cell wall (if required) or introduction of the first prey item is enough to cause anomalous airspeed readings as air flow in the pitot probe is impeded. Wasps have been observed inspecting aircraft noses within a few minutes of arrival at the gate, suggesting some experiential learning and memory of the nesting resource: the mobile and transient nature of the probes on aircraft (i.e. planes come and go) makes this choice of real pitot probes for nesting even more remarkable. The risk this species poses to aircraft at Brisbane Airport is significant, and management to reduce this risk includes covering of pitot probes when aircraft arrive and providing additional intercept traps to deter females from investigating pitot probes on aircraft [5].

*P. nasidens* is native to tropical South America, extending to the southern USA (Florida, Texas and Arizona), including islands in the Caribbean region, although it is possibly adventive in the USA [6]. It has been recorded from a number of Pacific islands including Hawaii and Micronesia. The spread of *P. nasidens* across the Pacific region does not follow a precise chronology [8] (recorded dates mark when the species was first observed) but does indicate that the species has been well-established outside its natural range since at least 1912. This dispersal is likely to have been through shipping, although the possibility that wasps are carried on aircraft cannot be discounted, especially as much of the spread is post WW2 when air traffic began to increase in the region: aircraft luggage bay temperatures may be sufficiently high (7˚C to over 25˚C: [32]) to allow adult wasps or pupae in nests to survive, and even wheel-wells may be suitable for shorter and lower altitude flights [33]. The appearance of *P. nasidens* in Hawaii as early as 1912 confirms that it arrived there by boat, as the first crossing of the Pacific to Hawaii (O'ahu) by aeroplane from the west coast of USA did not take place until 1927. The progression of the species across the Pacific is not a neat progression over time from east to west: this may reflect the opportunistic nature of dispersal, or simply the pattern of investigation of island invertebrate faunas by entomologists, or both.

As an adaptable, inventive and highly mobile species outside of its natural range, *P. nasidens* has the potential to spread from Brisbane to other locations in Australia where climates are suitable. Having arrived in Australia, the species has established in a challenging environment but one that provides all the basic requirements for population persistence and has identified a potential nesting opportunity that is both transient and mobile. In doing so, *P. nasidens* poses a significant risk to aviation safety [5], and further work is warranted to determine the prospects for its control or, preferably, eradication.

## Acknowledgments

We would like to thank Qantas (Matt Hill and Brian Ford) for their support of this project and their assistance with manufacturing the probes and panels. Matt has also been an extremely enthusiastic champion of the work and provided critical information on pitot probes and aviation operations. Dave Selby and Peter Dunlop (Brisbane Airport Corporation) were early supporters of this work, and Ross Wylie (Biosecurity Queensland) gave useful advice on wasp ecology. Chris Burwell (Queensland Museum) provided wasp identifications, Jeff McKee, Susan House and Carissa Free provided helpful comments on an earlier manuscript, and May-Le Ng kindly assisted with the analyses.

## Author Contributions

**Conceptualization:** Alan P. N. House, Jackson G. Ring.

**Data curation:** Alan P. N. House.

**Formal analysis:** Alan P. N. House.

**Funding acquisition:** Jackson G. Ring.

**Investigation:** Alan P. N. House, Phillip P. Shaw.

**Methodology:** Alan P. N. House, Phillip P. Shaw.

**Project administration:** Alan P. N. House.

**Resources:** Jackson G. Ring.

**Writing – original draft:** Alan P. N. House.

**Writing – review & editing:** Jackson G. Ring, Phillip P. Shaw.

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
