## [Decision Letter · Decision Letter 0]

5 Mar 2020

PONE-D-19-35094

Inventive nesting behaviour in the keyhole wasp *Pachodynerus nasidens* Latreille (Hymenoptera: Vespidae) in Australia, and the risk to aviation safety

PLOS ONE

Dear Alan House

Thank you for submitting your manuscript to PLOS ONE. After careful consideration, we feel that it has merit but does not fully meet PLOS ONE’s publication criteria as it currently stands. Therefore, we invite you to submit a revised version of the manuscript that addresses the points raised during the review process.

We would appreciate receiving your revised manuscript by one month. To enhance the reproducibility of your results, we recommend that if applicable you deposit your laboratory protocols in protocols.io, where a protocol can be assigned its own identifier (DOI) such that it can be cited independently in the future. For instructions see: http://journals.plos.org/plosone/s/submission-guidelines#loc-laboratory-protocols

We look forward to receiving your revised manuscript.

Kind regards,

Kleber Del-Claro, PhD

Academic Editor

PLOS ONE

Additional Editor Comments (if provided): Please, follow all reviewers suggestions

Journal Requirements:

4. Thank you for stating the following in the Financial Disclosure section:

"Brisbane Airport Corporation funded this work through contracts to Ecosure and Eco Logical Australia. BAC authorised the publication. BAC (through co-author J.G. Ring) contributed to the design and data collection, and made contributions to the manuscript."

We note that you received funding from a commercial source: Brisbane Airport Corporation

We note that one or more of the authors are employed by a commercial company: Eco Logical Australia, Brisbane Airport Corporation and Ecosure Pty Ltd.

6. We note that Figure 1 in your submission contain satellite images which may be copyrighted. All PLOS content is published under the Creative Commons Attribution License (CC BY 4.0), which means that the manuscript, images, and Supporting Information files will be freely available online, and any third party is permitted to access, download, copy, distribute, and use these materials in any way, even commercially, with proper attribution. For these reasons, we cannot publish previously copyrighted maps or satellite images created using proprietary data, such as Google software (Google Maps, Street View, and Earth). For more information, see our copyright guidelines: http://journals.plos.org/plosone/s/licenses-and-copyright.

Reviewers' comments:

Reviewer's Responses to Questions

**Comments to the Author**

1. Is the manuscript technically sound, and do the data support the conclusions?

Reviewer #1: Partly

Reviewer #2: Partly

2. Has the statistical analysis been performed appropriately and rigorously? 

Reviewer #1: No

Reviewer #2: No

3. Have the authors made all data underlying the findings in their manuscript fully available?

Reviewer #1: No

Reviewer #2: No

4. Is the manuscript presented in an intelligible fashion and written in standard English?

Reviewer #1: Yes

Reviewer #2: Yes

5. Review Comments to the Author

Reviewer #1: Dear Authors, I found your study very interesting and Ia also realize the urgency to publish this MS as it can help preventing accidents in airports.

I have made several comments and suggestions to enhance the quality of the MS, and I am especially concerned with the data and analyses performed. You have such rich data but they were not explored in its full magnitude. I also reommended several changes in the Methods section, to make the MS replicable.

Specific comments:

L 21 - 22 please rephase this sentence

Introducion - you need to work hard on your objectives and hypotheses (you can have hypotheses). As I read you results, I realized that you introduction (is good) lacks more robust objectives. You have investigated so many things, such as sex rations, influence of weather, environment and etc, and these factors are not presented in the Introduction.

L 57 - a parenthesis before Florida, but it does not close after Arizona (?)

L 163 - sample size = 12, correct? Or is there anymore treatments? How many probes per sheet? From your figures,

I counted 4 probes in one sheet and five probes in other sheet. Did you use all probe types (Table 1) in the experiment?

L 164 - incidents .... were suspected

Intercept traps - what is the sample size? For instance, you used 16 traps in airside and each trap consisted of 68 tubes, so in this

example you sample size is 68x16 = 1088 tubes? Is there another source of variation? Please provide these number in the MS, as it

helps to understand your statistical analyses. Moreover, you mention that tubes were replaced weekly, but it is not clear for how long.

Monitoring - here again I miss quantitative information.

"Probes were inspected weekly to monthly" - all probes? how many times were they inspected during the study period?

"Blocked probes were removed from the panels" - were these probes replaced by now ones?

Data Analysis - I think you should explore the data with the following tests. Please check them to see if they are appropriate to the data you collected.

I think using chi-squared tests are a poor indicator of statistical power.

lmm, glmm, or glmmTMB = number of occupied probes ~ location (4 locations) + probe size/volume/apperture + probe height + weather + time + (1|panels)

lm or Anova = occupied traps ~ location (2 locations) + time

L 196 - 201 - this shoud be on Methods

L 313. a dot is missing after P (wasp genus)

L 202 - I did not understand which variables you used in this multiple regression. What as the intercept, the number of wasps?

Results

L 208 - only here I see the duration of the study. I did not understand this number of 93 blocked probes. In the Methods section,

you need to provide more information on sample sizes. 93 probes correspond to what percentage of probes available? Here I see as well that there is

a time variable, and also a weather variable. From what you presented here, I cannot reverse the results to replicate a methodology.

Figure 3 - I suggest presenting the data in percent values, because you mention in the MS that you did not have a perfectly balanced design

L 231 - 233 - There must be problems with this Anova. I suggest changing the test (as I recommended above)

Figure 4 - these relations are not linear, and in Fig 4a there is an outlier in the data. Moreover, the text displays a value o R = 0.73, e the figure it is 0.53

Reviewer #2: This is an interesting study on the nesting ecology of a solitary wasp, Pachodynerus nasidens (Eumeninae). The authors made an innovative experiment based on a real problem frequently observed in airports. The study is important since provides the natural history aspects of the wasp and explores the preference of nesting sites which are the aircraft model pitots. The results are well explored but they used parametric ANOVA, so it is to state whether samples have normal distribution or may they transformed the raw data. Also, the data are not available.

The authors may suggest a further comment for companies to modify size or shape(?) of pitots since they showed a preference of probe aperture diameter between 4-7.2 mm.

6. PLOS authors have the option to publish the peer review history of their article (what does this mean?). If published, this will include your full peer review and any attached files.

Reviewer #1: No

Reviewer #2: No

---

## [Author Response · Author response to Decision Letter 0]

20 Apr 2020

Please see Responses to Reviewers file which addresses all comments from both reviewers. The updated covering letter includes responses to requests made by the journal, including Competing Interests and Funding Statements. We have elected not to submit a laboratory protocol - this was a field-based study, with all methods included in the manuscript. There were no laboratory analyses beyond those described in the MS..

---

## [Decision Letter · Decision Letter 1]

20 May 2020

PONE-D-19-35094R1

Inventive nesting behaviour in the keyhole wasp *Pachodynerus nasidens* Latreille (Hymenoptera: Vespidae) in Australia, and the risk to aviation safety

PLOS ONE

Dear Alan PN House

Thank you for submitting your manuscript to PLOS ONE. After careful consideration, we feel that it has merit but does not fully meet PLOS ONE’s publication criteria as it currently stands. Therefore, we invite you to submit a revised version of the manuscript that addresses the points raised during the review process.

We would appreciate receiving your revised manuscript by 20 days. To enhance the reproducibility of your results, we recommend that if applicable you deposit your laboratory protocols in protocols.io, where a protocol can be assigned its own identifier (DOI) such that it can be cited independently in the future. For instructions see: http://journals.plos.org/plosone/s/submission-guidelines#loc-laboratory-protocols

We look forward to receiving your revised manuscript.

Kind regards,

Kleber Del-Claro, PhD

Academic Editor

PLOS ONE

Additional Editor Comments (if provided):

Dear Author, our colleagues are delaying a lot in reviews mainly due to the crazy times we are living. Please, do all the modifications that the reviewer asked you and I will take the final decision without new round.

Best, wishes,

Kleber

Reviewers' comments:

Reviewer's Responses to Questions

**Comments to the Author**

1. If the authors have adequately addressed your comments raised in a previous round of review and you feel that this manuscript is now acceptable for publication, you may indicate that here to bypass the “Comments to the Author” section, enter your conflict of interest statement in the “Confidential to Editor” section, and submit your "Accept" recommendation.

Reviewer #1: All comments have been addressed

2. Is the manuscript technically sound, and do the data support the conclusions?

Reviewer #1: Yes

3. Has the statistical analysis been performed appropriately and rigorously? 

Reviewer #1: Yes

4. Have the authors made all data underlying the findings in their manuscript fully available?

Reviewer #1: No

5. Is the manuscript presented in an intelligible fashion and written in standard English?

Reviewer #1: Yes

6. Review Comments to the Author

Reviewer #1: Dear authors, I appreciate a lot the polite responses reagarding the first row of peer review. I still have some minor doubts/suggestions:

Please cite the R packages used in the study

How did you collecte the data of climatic variables?

Why including location as a random variable? I believe this is a fixed factor

L 234 - I did not understand the "Monte Carlo sample size". Is this misplaced?

L 244 - 249 and Table 2 - these are Methods (above) and not Data Analyses

L 261 - 264 - please prove the R² values

L 281 - The mixed effects model confirmed probe type was not significant but then you mention that they were significant.

This is a bit confusing. I suggest stating that probes were significant, because above you also show that probes > 5 mm were found to have nests

L 351 - 354 - are these r² values based on which test? Did you perform a regression here or is this part of the sex ratio test (Cochran-Armitage test). IF you are having difficulties to express the data collection and analyses, I suggest makibg a table like Table 1 (with you adaptations) made in the following papers. It helps a lot to understand the tests and figures associated with each objective.

1 - Coverdale, T. C., Goheen, J. R., Palmer, T. M., & Pringle, R. M. (2018). Good neighbors make good defenses: associational refuges reduce defense investment in African savanna plants. Ecology, 99(8), 1724-1736.

2 - Ancco Valdivia, F. G., Alves-Silva, E., & Del‐Claro, K. (2020). Differences in size and energy content affect the territorial status and mating success of a neotropical dragonfly. Austral Ecology.

Figure 4a - there is no need for this Figure

7. PLOS authors have the option to publish the peer review history of their article (what does this mean?). If published, this will include your full peer review and any attached files.

Reviewer #1: No

---

## [Author Response · Author response to Decision Letter 1]

18 Oct 2020

Responses to Reviewers

Reviewer #1: Dear authors, I appreciate a lot the polite responses regarding the first row of peer review. I still have some minor doubts/suggestions:

Please cite the R packages used in the study

Have updated citations to both R packages used.

How did you collect the data of climatic variables?

Detail and reference added.

Why including location as a random variable? I believe this is a fixed factor

We have retained location as a random variable as we had no a priori assumption or knowledge about where on the airport wasps were nesting.

L 234 - I did not understand the "Monte Carlo sample size". Is this misplaced?

Have changed sample size to number of simulations and added text – these are run as part of glmm to derive the most stable maximum likelihood results.

L 244 - 249 and Table 2 - these are Methods (above) and not Data Analyses

Text and table moved to Methods.

L 261 - 264 - please prove the R² values

Have changed these R values to adjusted R and added ANOVA results for proof.

L 281 - The mixed effects model confirmed probe type was not significant but then you mention that they were significant.

This is a bit confusing. I suggest stating that probes were significant, because above you also show that probes > 5 mm were found to have nests.

We agree this is bit confusing. The differences arise because for the formal test we could only use probe types that were deployed at all locations – this left out the smaller probes, so the result was not significant

L 351 - 354 - are these r² values based on which test? Did you perform a regression here or is this part of the sex ratio test (Cochran-Armitage test). 

Yes this is a result from the Cochran-Armitage test for trends. Have added text to indicate this.

IF you are having difficulties to express the data collection and analyses, I suggest making a table like Table 1 (with you adaptations) made in the following papers. It helps a lot to understand the tests and figures associated with each objective.

Have added Table 3. Thanks you for this suggestion!

Figure 4a - there is no need for this Figure

This figure has been removed

We have also made minor improvements to the text, also as tracked changes.

---

## [Decision Letter · Decision Letter 2]

27 Oct 2020

Inventive nesting behaviour in the keyhole wasp *Pachodynerus nasidens* Latreille (Hymenoptera: Vespidae) in Australia, and the risk to aviation safety

PONE-D-19-35094R2

Dear Dr. Alan PN House,

We’re pleased to inform you that your manuscript has been judged scientifically suitable for publication and will be formally accepted for publication once it meets all outstanding technical requirements.

Kind regards,

Kleber Del-Claro, PhD

Academic Editor

PLOS ONE

Additional Editor Comments (optional):

Reviewers' comments:

Reviewer's Responses to Questions

**Comments to the Author**

1. If the authors have adequately addressed your comments raised in a previous round of review and you feel that this manuscript is now acceptable for publication, you may indicate that here to bypass the “Comments to the Author” section, enter your conflict of interest statement in the “Confidential to Editor” section, and submit your "Accept" recommendation.

Reviewer #1: All comments have been addressed

2. Is the manuscript technically sound, and do the data support the conclusions?

Reviewer #1: Yes

3. Has the statistical analysis been performed appropriately and rigorously? 

Reviewer #1: Yes

4. Have the authors made all data underlying the findings in their manuscript fully available?

Reviewer #1: Yes

5. Is the manuscript presented in an intelligible fashion and written in standard English?

Reviewer #1: Yes

6. Review Comments to the Author

Reviewer #1: All suggestions were addressed by the authors. I believe the MS is resy for publication in its current form

7. PLOS authors have the option to publish the peer review history of their article (what does this mean?). If published, this will include your full peer review and any attached files.

Reviewer #1: No

---

## [Editor Report · Acceptance letter]

3 Nov 2020

PONE-D-19-35094R2 

Inventive nesting behaviour in the keyhole wasp *Pachodynerus nasidens* Latreille (Hymenoptera: Vespidae) in Australia, and the risk to aviation safety 

Dear Dr. House:

I'm pleased to inform you that your manuscript has been deemed suitable for publication in PLOS ONE. Congratulations! Your manuscript is now with our production department. 

Kind regards, 

on behalf of

Dr. Kleber Del-Claro 

Academic Editor

PLOS ONE